# Genome-Wide Identification and Expression Analysis of the Melon Aldehyde Dehydrogenase (ALDH) Gene Family in Response to Abiotic and Biotic Stresses

**DOI:** 10.3390/plants13202939

**Published:** 2024-10-21

**Authors:** Dekun Yang, Hongli Chen, Yu Zhang, Yan Wang, Yongqi Zhai, Gang Xu, Qiangqiang Ding, Mingxia Wang, Qi-an Zhang, Xiaomin Lu, Congsheng Yan

**Affiliations:** 1Institute of Vegetables, Anhui Academy of Agricultural Sciences, Hefei 230001, China18555104462@163.com (Y.Z.);; 2Key Laboratory of Genetic Improvement and Ecophysiology of Horticultural Crop, Hefei 230001, China; 3College of Agriculture, Anhui Science and Technology University, Fengyang 233100, China; 4Anhui Society for Horticultural Science, Hefei 230001, China

**Keywords:** ALDH gene family, melon, abiotic and biotic stress, RNA-seq

## Abstract

Through the integration of genomic information, transcriptome sequencing data, and bioinformatics methods, we conducted a comprehensive identification of the ALDH gene family in melon. We explored the impact of this gene family on melon growth, development, and their expression patterns in various tissues and under different stress conditions. Our study discovered a total of 17 ALDH genes spread across chromosomes 1, 2, 3, 4, 5, 7, 8, 11, and 12 in the melon genome. Through a phylogenetic analysis, these genes were classified into 10 distinct subfamilies. Notably, genes within the same subfamily exhibited consistent gene structures and conserved motifs. Our study discovered a pair of fragmental duplications within the melon ALDH gene. Furthermore, there was a noticeable collinearity relationship between the melon’s ALDH gene and that of *Arabidopsis* (12 times), and rice (3 times). Transcriptome data reanalysis revealed that some ALDH genes consistently expressed highly across all tissues and developmental stages, while others were tissue- or stage-specific. We analyzed the ALDH gene’s expression patterns under six stress types, namely salt, cold, waterlogged, powdery mildew, *Fusarium* wilt, and gummy stem blight. The results showed differential expression of *CmALDH2C4* and *CmALDH11A3* under all stress conditions, signifying their crucial roles in melon growth and stress response. RT-qPCR (quantitative reverse transcription PCR) analysis further corroborated these findings. This study paves the way for future genetic improvements in melon molecular breeding.

## 1. Introduction

Throughout their lifecycle, plants are subjected to various forms of abiotic and biological stresses. These stresses promptly induce the synthesis of reactive oxygen species (ROS) within the plants, subsequently leading to an accumulation of aldehydes [1]. Aldehydes, which are metabolic byproducts of carbohydrates, lipids, and amino acids, can, in appropriate quantities, positively influence the physiological metabolism of plants. However, excessive concentrations of aldehydes can cause cellular damage, disrupt metabolic processes, and lead to the formation of toxic substances detrimental to plant health, compromising their normal growth [2]. Hence, the selective removal and regulation of aldehyde concentrations within plants hold significant implications for their growth and development, mandating strict regulation [3].

Aldehyde dehydrogenase (ALDH), often referred to as the “aldehyde scavenger”, is effective in eliminating active aldehyde molecules in plants [4]. ALDH is a category of enzymes that encode NAD(P)+-dependent reactions, showcasing a diversity of amino acid sequences and incorporating different motifs such as the cysteine active site (PS00070), glutamic acid active site (PS00687), and the Rossmann fold [5]. To mitigate the damage inflicted by excessive aldehyde accumulation during plant growth, the ALDH gene is induced to express differently under stress, subsequently leading to a substantial increase in aldehyde dehydrogenase proteins [6]. Consequently, the toxic aldehydes generated due to stress are irreversibly oxidized into their non-toxic carboxylic acid counterparts, thus enhancing the plant’s stress tolerance and reducing stress-induced damage. This indicates a critical detoxification mechanism in plants [7]. Additionally, ALDH is crucial for various metabolic processes in plants, including glycolysis, amino acid synthesis, and the production of carnitine [8]. Under osmotic stress, ALDH helps maintain cellular osmotic equilibrium in plants through the synthesis of osmoregulators [9].

Aldehyde dehydrogenase (ALDH) is predominantly found in both eukaryotes and prokaryotes. As the genome sequencing information for an ever-increasing number of species becomes available, the presence of ALDH genes in more species is being discovered. The ALDH Gene Naming Committee (ANGC) defines a classification system based on protein sequence similarity between a new gene and an identified gene. If this similarity is less than 40%, the gene is considered as part of a new family; if the similarity ranges between 40% and 60%, it is classified within a family; if the similarity exceeds 60%, it is allocated to a subfamily [10]. Currently, the identified ALDH genes are categorized into 24 families, namely ALDH1-ALDH24, with 14 families found in plants. These 14 include ALDH2, ALDH3, ALDH5, ALDH6, ALDH7, ALDH10, ALDH11, ALDH12, ALDH18, ALDH19, ALDH21, ALDH22, ALDH23, and ALDH24, with the last seven exclusive to plants [11,12]. Over recent years, we have seen an increasing number of ALDH genes being identified in various plants, each with varying quantities, such as 14, 20, 28, 29, 53, 19, 27, 23, 22, and 39 in *Arabidopsis* [5], rice [13], maize [14], tomato [15], soybean [16], sorghum [17], Chinese cabbage [18], grape [19], potato [20], and apple [21], respectively. These genes play a significant role in the regulation of stress response, growth, and development in plants, hence are critical for normal plant growth [22,23]. For instance, *AtALDH3I1* and *AtALDH7B4* in *Arabidopsis* are seen to reduce MDA (malondialdehyde) and lipid peroxidation when their expression is upregulated, improving the plant’s tolerance to drought and salt stress [24]. Overexpression of *BrALDH7B2* in tobacco can enhance photosynthesis and reduce ROS production, thereby improving salt tolerance [25]. When wheat is subjected to drought and salt stress, *TraeALDH7B1-5A* is highly expressed in various parts of the plant, and overexpression of this gene in transgenic *Arabidopsis* significantly improves plant tolerance to drought stress [26]. Similarly, rice exhibited elevated expression levels of *OsALDH2*-4, *OsALDH3*-4, *OsALDH7*, *OsALDH18*-2, and *OsALDH12* when subjected to drought stress, surpassing control levels by twofold [13]. A Virus-Induced Gene Silencing (VIGS) experiment performed on the cotton genes *Gohir.A11G040800* and *Gohir.D06G046200* demonstrated increased sensitivity of the silenced plants to salt stress as compared to the control, indicating the potential involvement of these genes in cotton’s salt stress response [27]. In soybeans, acetaldehyde dehydrogenase can reduce the content of toxic MDA (malondialdehyde), and overexpression of *ALDH3H1* increases soybean resistance to salt stress [28]. *ZmALDH9*, *ZmALDH13*, and *ZmALDH17* have been reported to participate in the maize response to drought stress and are involved in plant acid tolerance and pathogen infection response [14].

*Cucumis melo* L., a member of the genus Cucumis in the Cucurbitaceae family, is an annual horticultural crop of high economic worth, primarily due to its flavorful fruits and substantial nutritional value [29,30,31]. In 2012, the genome of Muskmelon DHL92 was sequenced by the Spanish Agricultural Genomics Research Center, paving the way for bioinformatics research related to melon [32]. As bioinformatics technology continued to advance, Ruggieri carried out an upgrade of the DHL92 genome to version V3.6.1 in 2018, resulting in an enhanced sequence and annotation information [33]. Later in 2019, Castanera further improved the V3.6.1 version of DHL92 to version V4.0 [34]. Exploiting the melon genome data, a substantial volume of transcriptome sequencing and analysis has been executed by scholars [35]. This approach in transcriptomics research empowers the scientific community with a comprehensive understanding of relevant gene expression, allowing us to delve into the related metabolic networks and regulatory mechanisms at the transcription level [36]. The Cucurbitaceae genome database (CuGenDBv2, http://cucurbitgenomics.org/v2/, accessed 1 May 2024) has already made available a total of 41 published transcriptome data points related to melons. This wealth of data not only evidences the rapid progression in melon research but also facilitates the work of an increasingly large number of researchers. Through deep mining of the available transcriptome data, we can identify beneficial genes. In recent years, researchers worldwide have conducted numerous gene family studies, focusing on gene families such as LBD [37], SUN [38], and JMJ-C [39]. However, no research has yet been reported on the ALDH gene family in melon. In this bioinformatic analysis, we identified 17 ALDH genes in the melon genome using the available genomic data and conducted a thorough biogenic analysis, examining aspects such as chromosome localization, phylogeny, gene structure, and collinearity. To explore the expression patterns of the ALDH gene in different melon tissues under various stresses, we utilized published transcriptome data to analyze ALDH gene expression levels. The findings of this study establish a basis for future research on the ALDH gene family in melon and provide potential target candidate genes for resistance breeding in melon.

## 2. Results

### 2.1. Identification and Physicochemical Characteristics of the ALDH Family Members in Melon

This research identified 17 ALDH genes in the melon genome (DHL92V4) through the use of the hidden Markov model PF00171, and discovered conserved ALDH domains in Pfam, SMART, and NCBI conserved domain databases. An analysis of the physical and chemical properties of the ALDH family members in melon revealed that the smallest CDS gene was *CmALDH2B1* (1149 bp), while the largest was *CmALDH6B1* (3132 bp). The encoded amino acid length ranged from 382 (*CmALDH2B1*) to 1043 (*CmALDH6B1*); molecular weights fell between 41.81 kD (*CmALDH2B1*) and 114.38 kD (*CmALDH6B1*); and theoretical isoelectric point values varied from 5.32 (*CmALDH10A8*) to 8.89 (*CmALDH3H2*). Our instability index analysis indicated that only one ALDH gene in melon, *CmALDH6B1*, yielded a stable protein (instability coefficient > 40), while the rest were classified as unstable proteins (instability coefficient < 40). The aliphatic index (abundance and relative content of nonpolar amino acids in proteins) for the 17 ALDH genes spanned a range from 77.52 (*CmALDH6B1*) to 105.54 (*CmALDH18B2*). An evaluation of their average hydrophilic values revealed that 12 ALDH genes had discernible hydrophilic values (mean hydrophilic value < 0), with the remaining five classified as hydrophobic proteins. The mitochondria housed the most ALDH genes, with a total of six. This was followed by the chloroplast with five genes, the cytoplasm with four, and the nuclear and plasma membrane regions each containing one (Table 1).

### 2.2. Chromosome Distribution of Melon ALDH Genes

Utilizing the location information of ALDH gene family members on melon chromosomes, a distribution map of the ALDH gene in melon was constructed using TBtools software 2.136 (Figure 1). The findings revealed that ALDH genes were present on all chromosomes except for 6, 9, and 10, which lacked these genes, and a certain degree of variation in distribution quantity. For instance, the highest number of distributed genes was found on chromosomes 2 and 7, with three on each. Conversely, chromosomes 3, 4, 8, and 12 harbored the fewest genes, each with only one. This suggests that chromosomes 2 and 7 may play a significant role in the evolution of ALDH in melon. By observing gene positioning on the chromosomes, it was observed that ALDH genes in melon predominantly resided on the termini of chromosomes, with some located centrally. Furthermore, one pair of tandem repeat genes and fragment repeat genes, namely *CmALDH3H1/CmALDH3H2* and *CmALDH18B2/CmALDH18B1*, respectively, were identified within the 17 ALDH genes of melon.

### 2.3. Phylogenetic Tree Analysis of ALDH Family Genes

To elucidate the phylogenetic relationship of ALDH genes in melon, we procured the protein sequences of acknowledged ALDH genes from *Arabidopsis* and rice genome databases. These sequences were aligned with the 17 ALDH protein sequences in melon to facilitate phylogenetic analysis and construct a phylogenetic tree (Figure 2). Subsequently, melon’s ALDH genes were partitioned into 10 subfamilies based on *Arabidopsis* and rice groupings. Family 3 was identified as the most abundant, containing four melon ALDH genes, followed by Family 2 with three. Aside from Family 6 and 18, which each comprised two melon ALDH genes, the remaining six subfamilies each held a single ALDH gene.

We discovered seven pairs of direct homologous genes between the ALDH family of melon and *Arabidopsis*, namely *CmALDH11A3*/*AtALDH11A3*, *CmALDH10A8/AtALDH10A8*, *CmALDH2C4/AtALDH2C4*, *CmALDH2B2/AtALDH2B4*, *CmALDH12A1*/*AtALDH12A1*, *CmALDH6B1/AtALDH6B2*, and *CmALDH3F1/AtALDH3F1*. Additionally, we found one family of orthologous genes between melon and rice ALDH, specifically *CmALDH7B1/OsALDH7B6*. In the melon ALDH gene family, only one pair of paralog genes, *CmALDH3H2/CmALDH3H1*, was identified. Evolutionarily related ALDH genes, which exhibit similar biological functions and gene structures, were noted. Future studies can predict the biological function of the melon ALDH gene based on previously reported ALDH genes in *Arabidopsis* and rice.

### 2.4. Gene Structure and Conserved Motif Analysis of Melon ALDH Genes

We conducted a phylogenetic tree construction using the protein sequences from 17 melon ALDH genes, underpinned by gene structure and conserved motif analysis in correlation with melon genome files (Figure 3). The findings demonstrated that melon’s ALDH gene family could be segregated into 10 distinct subfamilies. ALDH genes originating from the same subfamily exhibited a tendency to cluster, which essentially aligns with the ALDH gene clustering outcomes from melon, *Arabidopsis*, and rice (Figure 2).

In the context of multi-gene families, the structural disparities in exon-introns within the same family serve as pivotal factors in the evolutionary process. As per the gene structure of ALDH genes in the melon family, the exon count fluctuates between 7 and 20. The genes *CmALDH5F1* and *CmALDH18B2* are characterized by the highest number of exons, tallying at 20, succeeded by *CmALDH6B2* and *CmALDH6B1* from the same subfamily, each having 19 exons. The lowest count is attributed to *CmALDH2B1*, with a mere 7 exons. Genes within the same subfamily generally exhibit similarities in their exon-intron structure. The resemblance in these gene structures, coupled with high sequence homology, suggests that gene duplication events transpired in the evolution of the melon’s ALDH gene family.

Conserved motif analysis revealed that the motif sequences of Families 3, 22, 11, 5, 10, and 2 were nearly identical, in the sequence of 3, 1, and 2. The motifs in Family 6 and Family 7 displayed the same sequence, 3, 1. Similarly, the motifs in Family 12 and Family 18 followed the same sequence, incorporating only motif 3. This suggests that ALDH genes within these similar subfamilies could potentially share similar biological functions.

### 2.5. Synteny Analysis of ALDH Genes Among Melon, Arabidopsis, and Rice

To deepen our understanding of the evolutionary details of ALDH genes in melon, we conducted a synteny analysis of these genes (Figure 4). Our findings revealed that, out of the 17 ALDH genes in melon, there was only one pair of repeated gene fragments, specifically *CmALDH18B2/CmALDH18B1*. Nine melon ALDH genes (*CmALDH6B2*, *CmALDH7B1*, *CmALDH2C4*, *CmALDH2B2*, *CmALDH18B2*, *CmALDH3H1*, *CmALDH3F1*, *CmALDH5F1*, *CmALDH22A1*) displayed twelve types of collinearity relationships with the twelve genes in *Arabidopsis* (*AtALDH2B7*, *AtALDH3H1*, *AtALDH7B4*, *AtALDH10A8*, *AtALDH5F1*, *AtALDH6B2*, *P5CS1*, *AtALDH22A1*, *AtALDH2C4*, *AtALDH10A9*, *P5CS2*, *AtALDH3F1*). We also discovered three collinearity relationships between three melon ALDH genes (*CmALDH7B1*, *CmALDH10A8*, *CmALDH18B2*) and three rice ALDH genes (*OsALDH18B2*, *OsALDH10A5*, *OsALDH7B6*). However, the remaining six ALDH genes in melon (*CmALDH3F2*, *CmALDH2B1*, *CmALDH11A3*, *CmALDH3H2*, *CmALDH6B1*, *CmALDH12A1*) exhibited no collinearity with those in Arabidopsis and rice. This indicates that these genes demonstrate a certain degree of conservation within the ALDH family.

### 2.6. Analysis of the Cis-Acting Elements in Melon ALDH Genes

Based on the cis-acting element analysis, we identified 12 distinct cis-acting elements in the promoter sequences of the melon ALDH genes (Figure 5A). Light-responsiveness elements constituted the majority, accounting for 45.6% of the total, and were uniformly distributed among all 17 ALDH genes in melon. The second most prevalent were the hormone-related cis-acting elements (abscisic acid responsiveness, MeJA-responsiveness, and salicylic acid responsiveness), contributing to 28.6% of the total, following photoresponsive elements (Figure 5B). Additionally, we examined elements related to stress response and endosperm expression. Our findings suggest that the ALDH gene could respond to a range of hormones and stress conditions. Thus, the identified response elements could significantly impact the stress response capability of the ALDH gene under stress conditions.

### 2.7. Tissue-Specific Expression Analysis of Melon ALDH Genes

Utilizing previously published transcriptome data from an array of melon tissues and fruit at different developmental stages, we were able to analyze the tissue-specific expression profiles of the melon’s ALDH gene family members during various stages of development (Figure 6). The histospecificity analysis of the ALDH gene family revealed high expression levels of *CmALDH6B2* and *CmALDH10A8* across all organs. Notably, *CmALDH3H2* is expressed specifically in male flowers and showed nearly an 8-fold greater expression level compared to other organs. This is similar to *CmALDH2C4* and *CmALDH18B1*, which presented specific expression only in the roots and male flowers. Conversely, *CmALDH18B2* and *CmALDH22A1* exhibited low expression in male flowers, but high expression in other organs. *CmALDH7B1* and *CmALDH2B1* were highly expressed in all organs except the ovary, where expression levels were significantly lower. The expression of *CmALDH5F1* and *CmALDH12A1* were more evenly distributed across all organs. Interestingly, *CmALDH11A3* displayed significantly higher expression levels in both male and female flowers than other organs, which suggests it may play a role in the formation of melon flower organs and fruits. Among the 17 ALDH genes, only *CmALDH6B1* showed no specific organ expression and maintained low expression levels, suggesting this gene is not involved in the growth and development of melon organs.

The expression profiles of ALDH family genes during different stages of fruit development showed that *CmALDH6B2* and *CmALDH10A8* were highly expressed at all four stages (growing, ripening, climacteric, and post-climacteric). This pattern, combined with their organ-specific expression, suggests these genes participate in melon organ development and have a role in fruit ripening and development. *CmALDH7B1*, *CmALDH3H1*, *CmALDH5F1*, and *CmALDH12A1* also displayed high expression levels in all four stages, second only to *CmALDH6B2* and *CmALDH10A8*. However, *CmALDH2C4*, *CmALDH18B1*, *CmALDH3F2*, and *CmALDH3F1* exhibited very low or no expression throughout the entire fruit development stage. *CmALDH3H2* and *CmALDH2B1* showed stage-specific expression during the ripening and growing stages, respectively, while *CmALDH2B2* was significantly more expressed during the growing stage than at other stages. No expression was observed during the climacteric and post-climacteric stages, suggesting this gene may be involved only in the growth and ripening of melon fruits.

### 2.8. Expression Patterns Analysis of Melon ALDH Genes Under Abiotic Stresses

To investigate the expression patterns of the ALDH gene in melon under varying abiotic stress conditions, we used published transcriptome sequencing data from melon studies conducted under salt, cold, and waterlogged stress. This allowed us to analyze ALDH gene expression levels under these stresses (Figure 7).

Under salt stress (Figure 7A), only five ALDH genes—*CmALDH22A1*, *CmALDH6B1*, *CmALDH3F2*, *CmALDH3F1*, and *CmALDH3H2*—were not differentially expressed compared to the control. Conversely, *CmALDH2B2*, *CmALDH2C4*, and *CmALDH11A3* showed significant downregulation in both salt-sensitive and salt-resistant materials. Other genes, particularly *CmALDH2B1* and *CmALDH7B1*, were significantly upregulated in these same materials.

In response to cold stress (Figure 7B), only *CmALDH11A3* was significantly upregulated in both cold-sensitive and resistant materials when compared with the control. *CmALDH2B1* showed a downregulation in sensitive materials and an upregulation in resistant materials, whereas *CmALDH2C4* exhibited an opposed pattern of upregulation in sensitive and downregulation in resistant materials. Other genes—*CmALDH2B2*, *CmALDH6B1*, *CmALDH22A1*, *CmALDH3F2*, and *CmALDH3F1*—were significantly downregulated in both types of materials. However, downregulated expressions of *CmALDH6B2*, *CmALDH12A1*, *CmALDH5F1*, and *CmALDH7B1* were only observed in sensitive materials.

When under waterlogging stress (Figure 7C), *CmALDH2C4* was significantly downregulated after 24 h of treatment, but showed a significant upregulation after 72 h. Both *CmALDH3F1* and *CmALDH11A3* showed significant downregulation in all four treatments. *CmALDH2B2*, *CmALDH3F2*, and *CmALDH3H2* did not present differential expression at the 6 h mark but demonstrated downregulation at 24, 48, and 72 h.

### 2.9. Expression Patterns Analysis of Melon ALDH Genes Under Biotic Stresses

To comprehend the expression patterns of melon under various biological stresses, we reanalyzed the published transcriptomic data of melon under powdery mildew, *Fusarium* wilt, and gummy stem blight stress. This reanalysis was performed in conjunction with the melon genome data (DHL92 V4) using TBtools software to generate expression heat maps (Figure 8).

During powdery mildew stress (Figure 8A), *CmALDH11A3* and *CmALDH2C4* exhibited identical differential expression patterns. Their expression levels significantly decreased and were downregulated after 24 h of treatment in the sensitive material, then increased significantly and were upregulated following 168 h of treatment. *CmALDH3F2* showed downregulation in sensitive materials but upregulation in resistant materials. *CmALDH2B1* demonstrated a conflux of both upregulated and downregulated expressions in resistant materials. Both *CmALDH22A1* and *CmALDH2B2* were significantly downregulated in both sensitive and resistant materials, while *CmALDH18B2* showed significant downregulation only in sensitive materials. Conversely, *CmALDH6B2*, *CmALDH7B1*, and *CmALDH10A8* were significantly downregulated in resistant materials.

Under *Fusarium* wilt stress (Figure 8B), compared with the control materials, *CmALDH18B2*, *CmALDH7B1*, and *CmALDH10A8* showed the highest expression and significantly upregulated in both sensitive and resistant materials. Similarly, *CmALDH18B1* was significantly upregulated in both sensitive and resistant materials. In contrast, *CmALDH2C4* and *CmALDH6B1* demonstrated significant downregulation in both types of materials. *CmALDH10A8*, *CmALDH12A1*, and *CmALDH3H1* were significantly upregulated in sensitive materials, while *CmALDH11A3* and *CmALDH3F1* were significantly downregulated. Notably, *CmALDH5F1* was only significantly upregulated in resistant materials.

Under gummy stem blight stress (Figure 8C), five ALDH genes—*CmALDH7B1*, *CmALDH3H1*, *CmALDH6B2*, *CmALDH2B1*, and *CmALDH12A1*—were significantly upregulated in both sensitive and resistant materials compared with control materials. Excluding *CmALDH3H1*, the expression levels of the other four ALDH genes showed significant changes in sensitive materials. *CmALDH2B2* was downregulated in sensitive materials yet significantly upregulated in resistant ones. Conversely, *CmALDH11A3* and *CmALDH18B2* were significantly downregulated in sensitive materials. *CmALDH3F2*, *CmALDH3F1*, and *CmALDH2C4* were significantly upregulated in resistant materials. Lastly, *CmALDH5F1* demonstrated a significant downregulation in resistant materials.

### 2.10. Regulation Patterns of Melon ALDH Genes Under Stresses

The expression profiles of the melon ALDH gene family under various stresses were utilized to summarize and label the stress response of the ALDH gene family, and a heat map of ALDH genes under differing stresses was created using TBtools software (Figure 9). As indicated in the figure, 17 ALDH genes in melon participate in the plant’s stress response to varying extents. Notably, the ALDH genes *CmALDH2C4* and *CmALDH11A3* exhibited significant differential expression in both abiotic and biotic stress scenarios. This indicates that these two genes are capable of initiating a positive response under stress conditions. Hence, they may serve as key candidate genes for future studies examining melon’s stress responses. Interestingly, while *CmALDH2B2* did not exhibit different expression under *Fusarium* wilt stress, it was significantly downregulated under all other five forms of stress. Some ALDH genes in melon, such as *CmALDH3H2*, only showed downregulation under waterlogging stress conditions. In the context of abiotic stress, most of the 17 ALDH genes were upregulated under salt stress, but downregulated under cold and waterlogging stress. Under biological stress, the majority of these ALDH genes were downregulated in response to powdery mildew stress but upregulated in response to *Fusarium* wilt and gummy stem blight stress. These findings highlight the differential expression of melon’s ALDH gene family under abiotic and biological stresses, providing a favorable starting point for further research into the molecular biological functions of the ALDH gene family in melons.

### 2.11. RT-qPCR Analysis of the Melon ALDH Gene Family

To evaluate the precision of our transcriptomic data analysis, we selected all genes from the *CmALDH* gene family. We assessed the impact of salt stress treatment (300 mM) at varying time intervals (0 h, 3 h, 6 h, 12 h, and 24 h) on their expression levels using the RT-qPCR method (Figure 10). In this investigation, the transcriptomic data was salt stress treatment after 24 h. According to our RT-qPCR test results, the expression patterns were generally consistent for the two sets of data we focused on. Genes such as *CmALDH3F1*, *CmALDH6B1*, *CmALDH6B2*, *CmALDH7B1*, *CmALDH12A1* and *CmALDH18B1*, peaked in expression after 6 h of NaCl treatment (with CK serving as the control), showing significant differences in comparison to the control. Conversely, other genes, including *CmALDH2B1*, *CmALDH2C4*, *CmALDH3F2*, *CmALDH3H1*, *CmALDH3H2*, *CmALDH10A8*, and *CmALDH18B2* exhibited their highest expression levels and significant differences at the 12 h mark. Of these, the gene *CmALDH2B2* displayed a substantial decrease in expression following stress treatment, reaching its lowest point after 9 h, before demonstrating an increasing trend. The expression level of *CmALDH2B1* showed a rising pattern post-stress, exhibiting significant differences compared to the control. This increase was particularly notable at the 12 h point, after which it started to decrease.

## 3. Discussion

Research into plant responses to both abiotic and biotic stresses has consistently attracted the attention of scientists. Stress has the potential to not only diminish the yield and quality of food crops [40] but also, in severe instances, result in plant death, thereby incurring substantial losses in agricultural production [41]. The ALDH enzyme, aside from its protective and detoxifying influence, also plays a crucial role in plant stress response [42]. The most prominently studied gene within the ALDH gene family is the betaine aldehyde dehydrogenase gene. This gene, belonging to the ALDH10 family, is a well-established osmoregulator and has demonstrated an effective response to adversity. It becomes upregulated in response to both drought and salt stress, enhancing plant resilience to these stresses [43]. As scientific and technological progress has allowed for the publication of an increasing amount of genome and transcriptome sequencing information, it has become particularly valuable to utilize this data to identify stress-related genes. This approach not only increases the usefulness of the existing data but also holds significant implications for related research [44]. Melon, the second cucurbit crop to have its whole genome sequenced (as early as 2012) [32], has yet to have any studies published on its ALDH gene family. This lack of information significantly hinders research into melon’s biological functions linked to ALDH genes. Therefore, this study aims to fill this gap by conducting a whole genome identification and expression pattern analysis of the ALDH gene family in melons. The results will provide a theoretical framework for future ALDH gene family studies in melons and contribute to the identification of advantageous genes for resistance breeding in melons.

In this study, we identified 17 ALDH genes in the melon genome using the latest genomic information and advanced bioinformatics analysis methods. These findings differ from the number of ALDH genes identified in other species, which may be due to variations in the number of ALDH gene family members across different species and distinct stages of evolution [45]. It could also be linked to the specific characteristics of the melon’s chromosomes and genome [46]. In our phylogenetic analysis, these 17 melon ALDH genes were classified into 10 subfamilies, each with a largely consistent gene structure and motif arrangement. Comparative analysis with the model plants *Arabidopsis* and rice revealed that there are seven pairs of directly homologous genes with *Arabidopsis*, and only one pair with rice’s ALDH gene family. This suggests potential similarities in biological functions, allowing for future melon studies to refer to the homologous genes in *Arabidopsis* and rice. Our gene duplication analysis within the melon’s ALDH gene family revealed a pair of tandem and fragment repeats, suggesting that the amplification of the melon’s ALDH gene was solely due to these repeats, a finding that aligns with reports on gene families in other plants [47].

Transcriptomics forms a critical component of genomics research. Bolstered by the ongoing advancements in high-throughput sequencing technology, transcriptome sequencing is becoming increasingly prevalent in plant-related studies. This has led to deepened insights into gene expression regulatory mechanisms and cellular functional diversity in plants [48,49]. Numerous transcriptome sequencing projects have used melons as their subject, with the resulting data undergoing validation via RT-qPCR and rigorous peer review before publication [50,51]. This process has led to an extensive database of melon transcriptomes. This study leverages this wealth of published transcriptome sequencing data to analyze the expression patterns of 17 ALDH genes in melons across different tissues, developmental stages, and stress conditions.

ALDH’s role in plant organ development has been widely reported. For instance, the *ALDH2B2* (*Rf2a*) gene is linked to cytoplasmic male sterility in maize and plays a key role in another development [52]. ALDH is also highly expressed in rice anthers during early meiosis to microspore [53]. The *ALDH3F1* mutant triggers early flowering in plants, while its overexpression causes delayed flowering [54]. In grapes, the expression levels of three genes (*ALDH2B8*, *ALDH3H5*, and *ALDH18B1*) noticeably increase during fruit development and maturation, while the levels of *ALDH2B4* and *ALDH5F1* significantly decrease [11]. In this study, we found that all 17 ALDH genes in melon were expressed to varying degrees in different melon tissues, with some genes exhibiting tissue-specific expression. This specific expression pattern across various melon tissues contributes to the synergistic regulation of melon growth and development. Only four ALDH genes were not expressed at different developmental stages of melon fruit. Notably, the *CmALDH2B2* gene exhibited the highest expression level during the fruit growth stage and in male and female flowers and the ovary. However, its expression level dropped dramatically as the fruit matured, indicating that this gene primarily contributes to melon fruit growth.

The ALDH gene family’s response to stress in plants has become a key focus of recent studies, suggesting that this gene family is crucial for plant stress resistance. In European sweet cherry, fluorescence quantitative experiments confirmed that *PaALDH17*’s expression under salt stress was significantly higher than that of other genes. *PaALDH17* was subsequently cloned and transferred to *Arabidopsis*, where transgenic *PaALDH17* plants exhibited stronger stress resistance and fewer changes in various physiological indices under salt stress, compared to wild-type plants [55]. In grapes, *VvALDH2B4* and *VvALDH2B8* expression levels significantly increased under drought and salt stress [11]. Moreover, overexpressing *VvALDH2B4* in wild Chinese grapes led to a decrease in MDA levels, significantly improving resistance to pathogenic bacteria [56]. In bamboo, ALDH gene expression significantly increased under drought stress compared to the control, with *PeALDH2B2* responding to drought stress by interacting with *PeGPB1* [57]. Wang et al. [16] discovered that the wheat ALDH gene *TraeALDH7B1-5A* exhibited significant differences under both drought and salt stress. Its soybean homologous gene, *GmALDH7B1*, was significantly upregulated in both roots and leaves under 20% PEG treatment. To understand the melon ALDH gene family’s role in abiotic stress, this study analyzed the ALDH family genes’ expression patterns under salt, cold, and waterlogging stress using published melon transcriptome sequencing data. It was revealed that over half of the melon ALDH genes displayed significantly upregulated expression under salt stress, in line with previous reports on plant ALDH genes’ upregulated expression under salt and drought stress [20]. In contrast, most melon ALDH genes’ expression was significantly downregulated under cold and waterlogging stress. This suggests that these ALDH genes are negatively regulated when transcribed into mRNA under stress, leading to reduced expression [58]. Interestingly, some melon ALDH genes were differentially expressed under all three abiotic stresses, while others were only differentially expressed under one stress. For instance, *CmALDH10A8*, *CmALDH18B1*, *CmALDH3H1*, and *CmALDH18B2* were differentially expressed solely under salt stress, while their expression levels remained unchanged under cold and waterlogged stress. *CmALDH22A1* and *CmALDH6B1* were differentially expressed only under cold stress, and *CmALDH3H2* was differentially expressed only under waterlogging stress. These results suggest that while some ALDH genes can be induced under various abiotic stresses, others are only induced by a specific abiotic stress. In addition to abiotic stress analysis, this study also examined the ALDH gene family’s expression patterns in melons under biological stress. It was observed that the 17 melon ALDH genes were primarily downregulated under powdery mildew stress and primarily upregulated under *Fusarium* wilt stress, with a higher number of differentially expressed genes than under powdery mildew stress. These ALDH genes were positively responsive to these stresses and played a role in plant disease resistance [59]. Among these, *CmALDH22A1* was significantly downregulated under powdery mildew stress but not expressed under *Fusarium* wilt stress. Similarly, *CmALDH6B1* and *CmALDH18B1* were differentially expressed under *Fusarium* wilt stress. However, *CmALDH3H2* was not differentially expressed under any of these three biological stresses, suggesting that this gene did not contribute to melon’s response to these biological stresses. The functional characteristics of melon ALDH genes showed that 17 ALDH genes were differentially expressed under both abiotic and biological stresses, with *CmALDH2C4* and *CmALDH11A3* being differentially expressed under all stresses (salt, cold, waterlogging, powdery mildew, *Fusarium* wilt, and gummy stem blight). Interestingly, *CmALDH2C4* was upregulated and downregulated in cold, waterlogged, and powdery mildew stress, suggesting that these two melon ALDH genes play a role in plants facing these stresses.

## 4. Materials and Methods

### 4.1. Identification and Chromosomal Distribution of Melon ALDH Gene Family Members

We retrieved the melon DHL92 protein group sequence file from the Cucurbitaceae genome database (http://cucurbitgenomics.org/v2/, accessed 5 May 2024), and constructed a local database [60]. Subsequently, the HMM model file for the ALDH gene family (PF00171) was downloaded from the InterPro database (https://www.ebi.ac.uk/interpro/, accessed 5 May 2024) [61]. We utilized the hmmsearch program [62] to screen the ALDH gene ID (E < 1 × 10^5^) in the melon protein database, and a Perl script was employed to extract the protein sequence data of the candidate genes. These sequences were then uploaded to the online platforms Pfan [63], SMART [64], and NCBI [65] for domain verification of identified ALDH gene candidates in melon. Sequences that did not contain the ALDH domain were eliminated, aiding in the identification of ALDH gene family members in melon. The ExPASy online tool [66] (https://web.expasy.org/protparam/, accessed 5 May 2024) was used to analyze the physical and chemical properties of the melon ALDH family genetic information. Lastly, the chromosome distribution map of the ALDH family genes in melon was constructed using the TBtools software [67].

### 4.2. Phylogenetic Analysis of ALDH Family Genes from Melon, Arabidopsis, and Rice

The sequence data for the ALDH gene family members in melon were uploaded to the online platform, GSDS (http://gsds.cbi.pku.edu.cn/, accessed 8 May 2024) [68], to execute a gene structure analysis. For the examination of conserved motifs within the ALDH gene family, the same sequence information was submitted to the online resource, MEME (http://meme-suite.org/tools/meme, accessed 8 May 2024) [69], with all parameters set to their default values. In order to construct a phylogenetic tree, we combined ALDH protein sequences of *Arabidopsis* and rice with those of melon. This collective dataset was then imported into the MEGA11 software (version 11.0.13) [70]. The construction process employed the maximum likelihood method, with all parameters set to their default values.

### 4.3. Genetic Characterization and Phylogenetic Analysis of ALDH Family in Melon

The MCScanX software (using built-in parameters) [71] was employed to examine tandem and fragment duplications within the ALDH gene family in melon. Furthermore, it was used to investigate the collinearity relationship among the ALDH gene families in *Arabidopsis*, rice, and melon. The findings were then visualized using the TBtools software. The cis-acting elements of the promoters of the melon ALDH family genes were analyzed within the 2.0 kb upstream sequences from the transcription start sites of the melon ALDH family genes using the online website PlantCare (https://bioinformatics.psb.ugent.be/webtools/plantcare/html/, accessed 8 May 2024).

### 4.4. Tissue-Specific Expression of Melon ALDH Family Genes

We reanalyzed publicly available melon transcriptome data from different tissues (PRJNA803327) [30] and developmental stages (PRJNA543288) [72] in conjunction with melon genome information. Subsequently, TBtools was utilized to generate a heatmap representation of ALDH gene family expression in various melon tissues and at different stages.

### 4.5. Analysis of Expression Patterns of Melon ALDH Gene Family Under Various Stresses

In order to investigate the effect of different stresses, we evaluated published melon transcriptome data under conditions including salt stress (PRJNA296827) [73], cold stress (PRJNA553119) [74], waterlogging stress (PRJNA726294) [75], powdery mildew (PRJNA358655) [76], *Fusarium* wilt (PRJNA842515) [77], and gummy stem blight (PRJNA681992) [78]. This data was reanalyzed in combination with melon genomic data, enabling us to identify differentially expressed genes. A heatmap of their expression under different stress conditions in melon was again visualized using TBtools.

### 4.6. RT-qPCR Analysis of Melon Under Salt Stress

We used “TQ1”, a highly inbred melon strain, provided by the Institute of Vegetables, Anhui Academy of Agricultural Sciences, as the experimental subject. Melon seedlings, possessing uniform growth potential and exhibiting five cotyledon leaves and one true leaf, were exposed to a 300 mM NaCl stress for periods of 0, 3, 6, 9, 12, and 24 h. Subsequently, the young leaves were harvested for further analysis. The leaves were then flash-frozen in liquid nitrogen for RNA extraction and reverse transcription. Total RNA extraction was performed utilizing the HiPure Plant RNA Mini Kit (Magen Biotech, Shanghai, China), followed by cDNA synthesis via the SMART kit (Takara Bio, Shiga, Japan), all in strict adherence to the manufacturer’s guidelines. The real-time quantitative RT-PCR (RT-qPCR) was conducted employing SYBR Green qPCR Premix (Low ROX) on an IQ5 real-time PCR detection system (Bio-Rad, Hercules, CA, USA). *Actin3* (GenBank accession number XM_008449644.2) was utilized as the reference gene (Appendix A). The reaction system consisted of the following: ddH2O (7 μL), 2 × Mix (10 μL), cDNA (1 μL), and both positive and negative primers (1 μL each), the reaction system was 20 μL and the reaction procedure was 95 °C for 30 s, 95 °C for 5 s, 60 °C for 34 s, 40 cycles; 72 °C for 10 s. The results from three biological replicates were analyzed using the 2^−ΔΔCt^ method and processed with Excel 2021. A *t*-test in SPSS 19.0 was employed to assess the significance of differences in the data, and GraphPad 9.0 Prism was utilized for plotting.

## 5. Conclusions

In this research, we undertook the first-ever identification of the ALDH gene family within the melon genome. We identified a total of 17 ALDH gene members, which were scattered across chromosomes 1, 2, 3, 4, 5, 7, 8, 11, and 12. Phylogenetic analysis grouped these genes into 10 distinct subfamilies, each exhibiting a largely similar gene structure. Collinearity analysis revealed a pair of fragment repeats among the 17 melon ALDH genes. Additionally, there were 12 collinear relationships between melon and *Arabidopsis* and three collinear relationships between melon and rice. Analysis of ALDH gene expression patterns during different developmental stages and in various melon tissues suggested that these genes contribute to melon growth, development, and fruit ripening. The examination of ALDH gene expression patterns under diverse stress conditions showed that these genes were differentially expressed at varying degrees in response to different stresses in melons. Particularly, the distinct expressions of *CmALDH2C4* and *CmALDH11A3* signpost these genes’ crucial role in melon growth and development. RT-qPCR analysis results showed that most genes were highly expressed, signifying their participation in salt stress responses. Together, these findings pave the way for further exploration of the ALDH gene family in melons and offer promising genes for enhancing melon resistance breeding.

## Figures and Tables

**Figure 1 plants-13-02939-f001:**
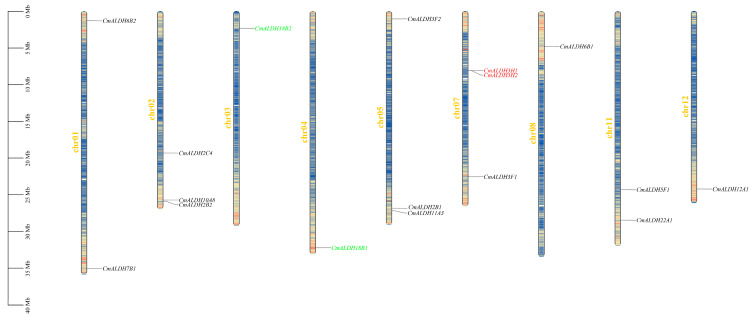
Distribution of the melon ALDH family genes on the chromosomes. The genes marked in red color were tandem duplication gene pairs, and the genes marked in green color were segmental duplication genes. The blue part on the chromosome indicates a small number of distributed genes, the yellow part indicates a moderate number of distributed genes, while the red part indicates a large number of distributed genes.

**Figure 2 plants-13-02939-f002:**
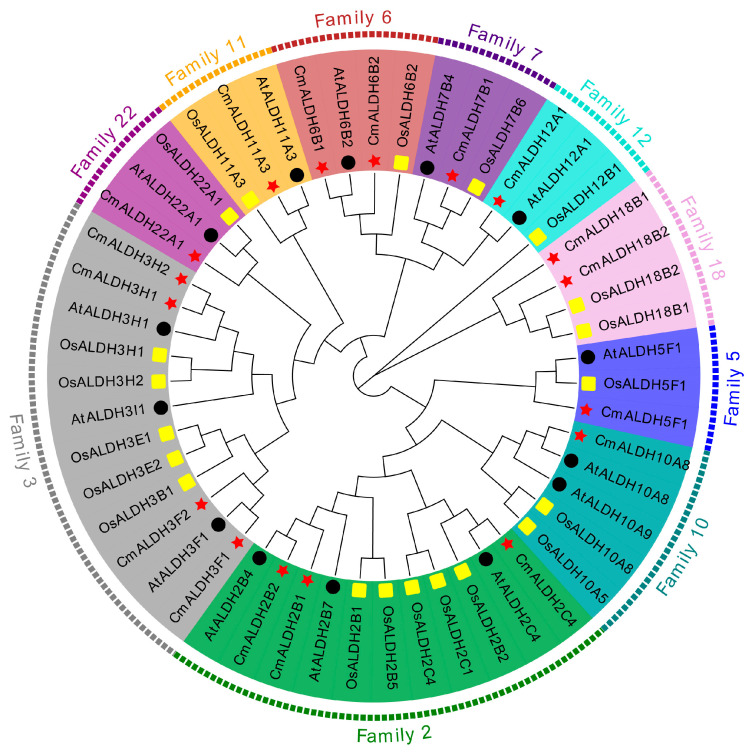
Phylogenetic relationships of melon, rice, and *Arabidopsis* ALDH proteins. The red star represents the protein in melon. The black circle represents the protein in *Arabidopsis*. The yellow rectangle represents the protein in rice.

**Figure 3 plants-13-02939-f003:**
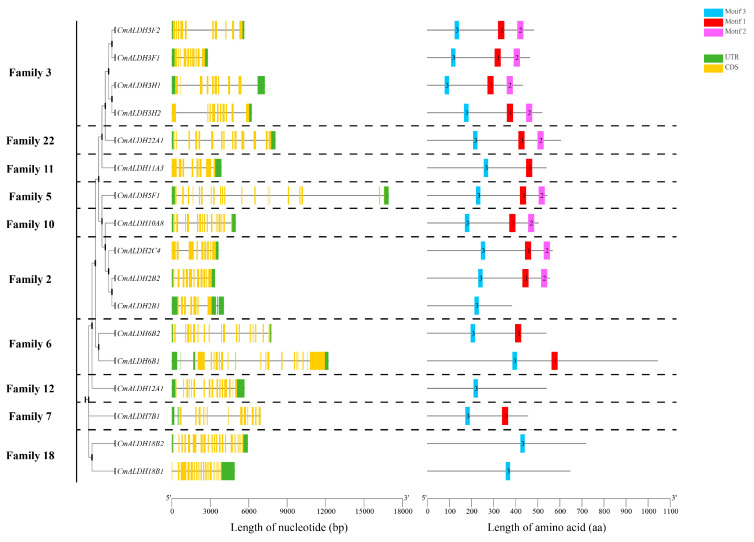
Gene structures and protein motifs of the melon ALDH gene family.

**Figure 4 plants-13-02939-f004:**
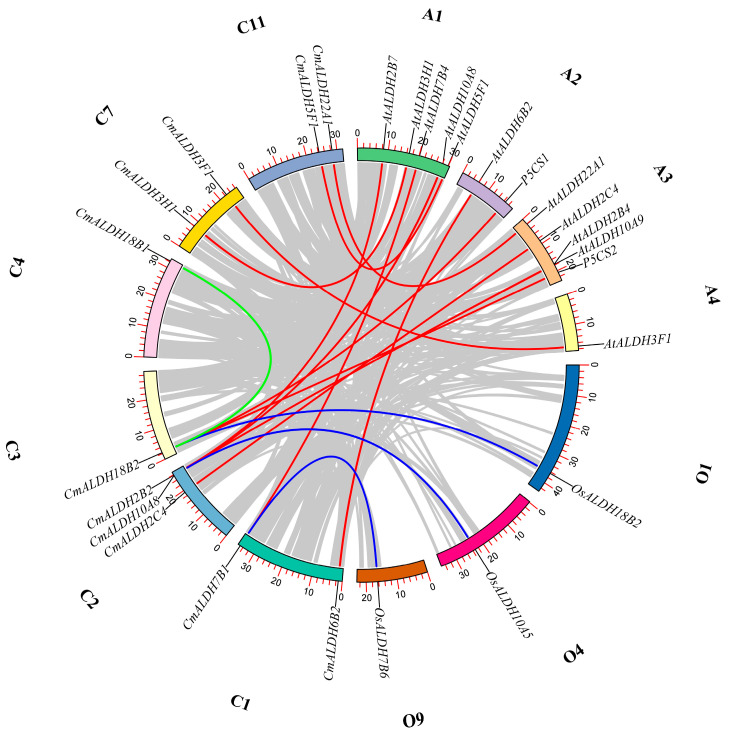
Assessment of the ALDH gene duplication between melon, *Arabidopsis*, and rice. The green lines represent the segmentally duplicated ALDH genes in melon. The red lines represent the orthologous relationships of the ALDH genes between melon and *Arabidopsis*. The blue lines represent the orthologous relationships of the ALDH genes between melon and rice. The gray lines represent the orthologous relationships of the genes between melon, *Arabidopsis*, and rice. The scale on the chromosome represents the physical location.

**Figure 5 plants-13-02939-f005:**
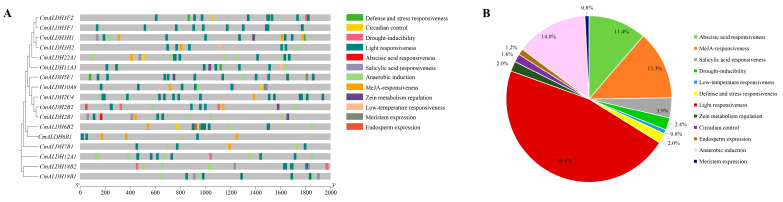
*Cis*-regulatory elements in promoter region of melon ALDH genes (**A**) and their proportions (**B**).

**Figure 6 plants-13-02939-f006:**
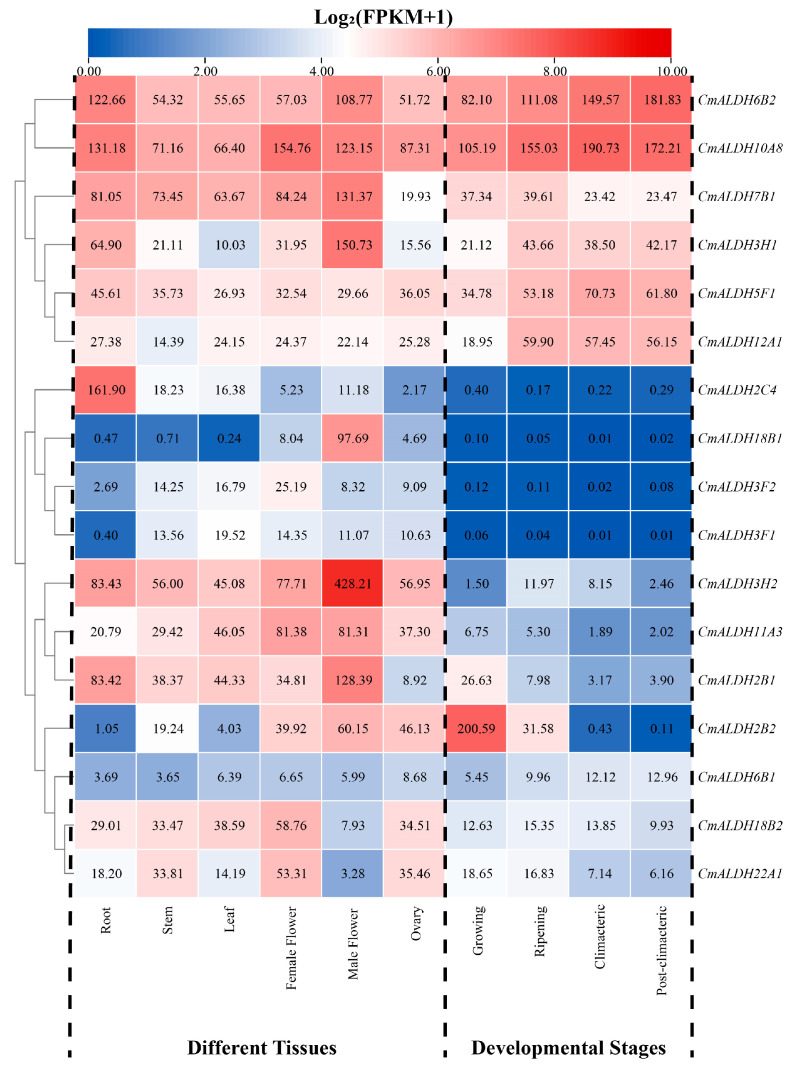
Expression analysis of melon ALDH genes in different tissues and developmental stages.

**Figure 7 plants-13-02939-f007:**
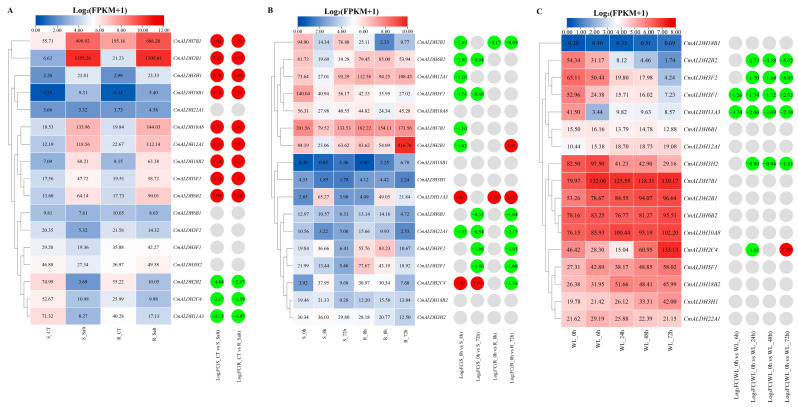
The expression patterns of melon ALDH genes in response to abiotic stresses. (**A**) The expression heatmap of melon ALDH genes in response to salt stress. S: susceptible plant; R: resistant plant; CT: control treatment; Salt: salt treatment. (**B**) The expression heatmap of melon ALDH genes in response to cold stress. S: susceptible plant; R: resistant plant; 0 h, 8 h and 72 h were treatment for 0 h, 8 h and 72 h. (**C**) The expression heatmap of melon ALDH genes in response to waterlogging stress. 0 h, 6 h, 24 h, 48 h and 72 h were treatment for 0 h, 6 h, 24 h, 48 h and 72 h hours. The data in the left expression heatmaps are the original FPKM values; the data in the right boxes are log2 (fold change) values highlighted by red (upregulation) and green (downregulation) colors.

**Figure 8 plants-13-02939-f008:**
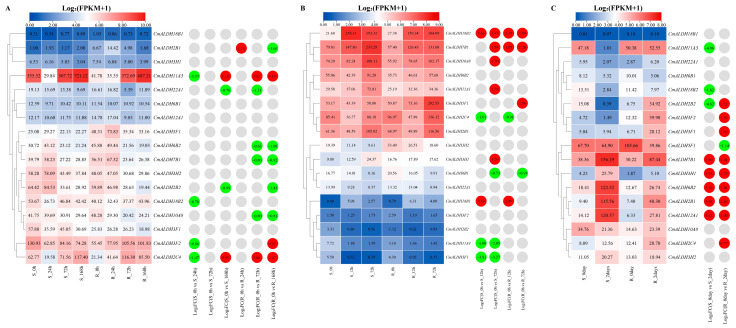
The expression patterns of melon ALDH genes in response to biotic stresses. (**A**) The expression heatmap of melon ALDH genes in response to downy mildew stress. S: susceptible plant; R: resistant plant; 0 h, 24 h, 72 h and 168 h were 0, 24, 72, and 168 h post-inoculation, respectively. (**B**) The expression heatmap of melon ALDH genes in response to *Fusarium* wilt stress. S: susceptible plant; R: resistant plant; 0 h, 12 h, and 72 h were 0, 12, and 72 h post-inoculation, respectively. (**C**) The expression heatmap of melon ALDH genes in response to gummy stem blight stress. S: susceptible plant; R: resistant plant; 0 day and 2 days were 0 and 2 days post-inoculation, respectively. The data in the left expression heatmaps are the original FPKM values; the data in the right boxes are log2 (fold change) values highlighted by red (upregulation) and green (downregulation) colors.

**Figure 9 plants-13-02939-f009:**
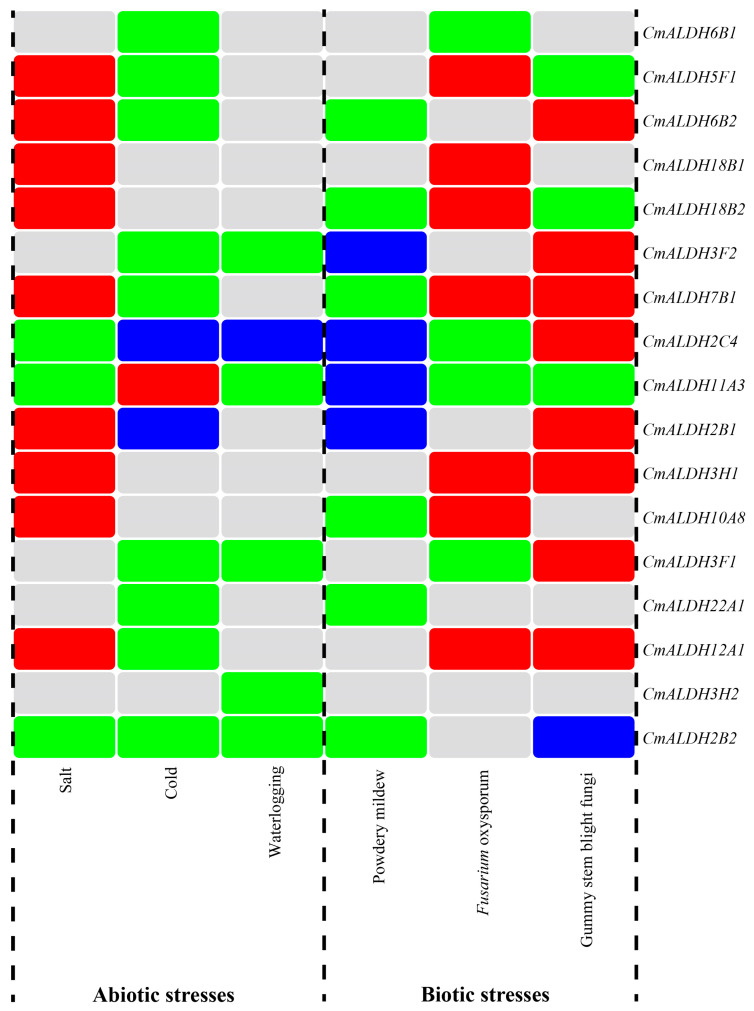
An expression pattern heatmap of the melon ALDH genes under abiotic and biotic stresses. The gray color represents unchanged expression, red represents upregulated expression, green represents downregulated expression, and blue represents both upregulated and downregulated expression.

**Figure 10 plants-13-02939-f010:**
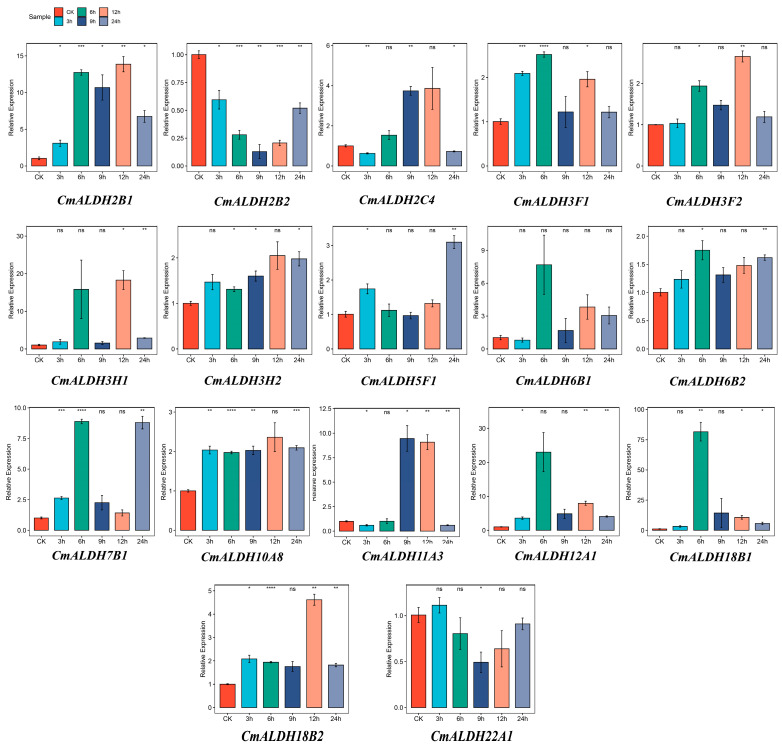
Relative expression level of 17 *CmALDH* genes in response to salt treatments. Error bars are the average error of three biological replicates. Asterisks are used to indicate the significant degree of the expression level compared to the value of the control (* *p* <  0.05, ** *p* < 0.01, *** *p* < 0.001, **** *p* < 0.0001).

**Table 1 plants-13-02939-t001:** Information and physicochemical properties of 17 *CmALDH* members.

Gene Name	Locus Name	CDS Size (bp)	Number of Amino Acids (aa)	Molecular Weight (kDa)	TheoreticalpI	Instability Index	Aliphatic Index	Grand Average of Hydropathicity	Prediction of Subcellular Location
*CmALDH2B1*	*MELO3C004383*	1149	382	41.81	8.30	26.07	93.19	−0.022	Chloroplast
*CmALDH2B2*	*MELO3C017100*	1665	554	60.07	7.23	33.22	89.10	−0.084	Chloroplast
*CmALDH2C4*	*MELO3C025328*	1701	566	61.73	8.81	31.75	87.05	−0.069	Mitochondrial
*CmALDH3F1*	*MELO3C017542*	1395	464	51.87	8.76	37.35	97.93	−0.06	Cytoplasmic
*CmALDH3F2*	*MELO3C014601*	1446	481	54.04	8.55	39.02	95.90	−0.043	Plasma Membrane
*CmALDH3H1*	*MELO3C010494*	1296	431	47.24	8.11	34.49	104.29	0.046	Cytoplasmic
*CmALDH3H2*	*MELO3C010493*	1560	519	57.19	8.89	35.71	99.79	0.02	Mitochondrial
*CmALDH5F1*	*MELO3C019622*	1626	541	57.94	6.98	38.09	95.21	0.039	Chloroplast
*CmALDH6B1*	*MELO3C007705*	3132	1043	114.38	8.10	48.04	77.52	−0.38	Nuclear
*CmALDH6B2*	*MELO3C018583*	1614	537	57.55	6.47	34.29	90.63	0.041	Mitochondrial
*CmALDH7B1*	*MELO3C024345*	1365	454	48.7	6.19	37.34	96.17	0.085	Chloroplast
*CmALDH10A8*	*MELO3C017125*	1512	503	54.56	5.32	25.17	92.94	−0.001	Cytoplasmic
*CmALDH11A3*	*MELO3C004430*	1620	539	58.48	7.60	37.92	91.37	−0.002	Chloroplast
*CmALDH12A1*	*MELO3C002203*	1620	539	60.37	6.33	39.65	95.45	−0.09	Mitochondrial
*CmALDH18B1*	*MELO3C009229*	1941	646	69.86	5.45	34.89	104.78	−0.032	Cytoplasmic
*CmALDH18B2*	*MELO3C008245*	2154	717	77.67	6.20	32.92	105.54	−0.096	Mitochondrial
*CmALDH22A1*	*MELO3C021272*	1815	604	66.88	7.20	36.64	92.33	−0.032	Mitochondrial

## Data Availability

The data presented in this study are available on request from the corresponding author. Congsheng Yan (congshengyan@126.com).

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
