# Peer review of "Genome-Wide Identification and Expression Analysis of the Melon Aldehyde Dehydrogenase (ALDH) Gene Family in Response to Abiotic and Biotic Stresses"

_plants, 2024, doi:10.3390/plants13202939_

Round 1
Reviewer 1 Report
Comments and Suggestions for Authors
TITLE:
The full name of ALDH should be provided in the title.
ABSTRACT:
Line 22, fragmental duplication
Line 28, I did not find the MELO3C024328 gene in Table 1
Line 29, full name of RT-qPCR
INTRODUCTION:
Since this study is heavily relied on previous published transcriptome dataset, I suggest that they should comprehensively introduce the achievement of these studies, helping the readers understand the advancements and novelty of the current study.
Several parts of the content are highly similar to Islam and Ghosh (2022, Gene). The authors should recapitulate the information based on their own understanding and knowldges.
Line 76 and 89, MDA
Line 105, bioinformatic analysis
RESULTS:
Section 2.1, I suggest that the author consider renaming the melon ALDH genes, such as ALDH1, etc. in this way, it would be feasible for the readers focusing on the selected candidates.
Line 119, aa, amino acid.
Line 120, 41.8 kDa. Corrections should be made in the Table 1, accordingly.
Line 125, what does the fat co-efficiency indicate?
Line 163, melon or muskmelon?
Figure 2, the legend is incomplete. What does the star, black dots, etc. indicate?
Figure 3, the background color is too dark to read the GeneID. The signature motifs of ALDH family, such as the ones that introduced in the Introduction, should be marked in the right panel. The legend needs to be improved.
Line 204-210, I don’t understand.
Figure 4, the legend need to be improved. What does the gray line indicate? What does the scale on the chromosome mean?
Section 2.6, the authors selected 2 kb upstream of CmALDH gene to conduct the promoter analysis? This relevant information should be provided.
Figure 6 to 8, the statistic analysis results and the raw data for the heat maps should be provided. The value inside each brick is not inconsistent with the color code.
Line 284, chilling stress.
Section 2.11, which standard did the authors apply to select the 10 candidates for RT-qPCR analysis? I suggest examining the expression changes of all of the candidates.
DISCUSSION:
Line 406, biotic stress
Line 481 to 508, the authors recite the Results without any insightful statements. I suggest that they should integrate the results of tissue-specific, developmental and stress responsive transcriptome analysis to conclude the functional model of selected candidate.
Line 511, MELO3C024328?
M&M:
Section 4.6, the methodology of RT-qPCR is incomplete. Such as the chemical regents/kits used for RNA extraction and reverse transcription. Which gene was used as reference gene?
Methodology of statistic analysis should be provided.
Comments on the Quality of English Language
Minor editing of English language required.
Author Response
Comments1: The full name of ALDH should be provided in the title.
Response 1: Thank you for pointing this out. We agree with this comment. Therefore, we have changed "ALDH" to "Aldehyde dehydrogenase (ALDH)", this change can be seen on line 3.
Comments2: Line 22, fragmental duplication
Response 2: Agree. We have changed “fragment repeats” to “fragmental duplication”.
Comments3: Line 28, I did not find the MELO3C024328 gene in Table 1
Response 3: Thank you for the comment. We have corrected the "MELO3C024328 "to "MELO3C025328 " in the revised manuscript.
Comments4: Line 29, full name of RT-qPCR
Response 4: We sincerely thank your precious suggestion. We have corrected the "RT-qPCR"to "RT-qPCR (quantitative reverse transcription PCR)" in the revised manuscript.
Comments5: Since this study is heavily relied on previous published transcriptome dataset, I suggest that they should comprehensively introduce the achievement of these studies, helping the readers understand the advancements and novelty of the current study.
Response 5: Thank you for the comment. We have made a detailed introduction to transcriptome-related research according to your suggestion in the introduction section.
Comments6: Several parts of the content are highly similar to Islam and Ghosh (2022, Gene). The authors should recapitulate the information based on their own understanding and knowldges.
Response 6: We sincerely thank your precious suggestion. We have revised this text in accordance with your opinion as follows: Additionally, ALDH is crucial for various metabolic processes in plants, including glycolysis, amino acid synthesis, and the production of carnitine.
Comments 7:Line 76 and 89, MDA.
Response 7: We sincerely thank your precious suggestion. We have unified it as “MDA (malondialdehyde)”.
Comments8: Line 105, bioinformatic analysis.
Response 8: We sincerely thank your precious suggestion. We have corrected the "In this study "to "In this bioinformatic analysis" in the revised manuscript.
Comments 9:Section 2.1, I suggest that the author consider renaming the melon ALDH genes, such as ALDH1, etc. in this way, it would be feasible for the readers focusing on the selected candidates.
Response 9: Thank you for the comment. We have renamed the genes according to your suggestion. See Table 1 for details.
Comments 10:Line 119, aa, amino acid.
Response 10: We sincerely thank your precious suggestion. We have made corresponding modifications.
Comments11: Line 120, 41.8 kDa. Corrections should be made in the Table 1, accordingly.
Response 11: We sincerely thank your precious suggestion. We have corrected the relevant errors in the manuscript and Table 1.
Comments 12:Line 125, what does the fat co-efficiency indicate?
Response 12: Thank you for the comment. This was our mistake. It should be "aliphatic index", not " fat co-efficiency". We have made corresponding modifications in the manuscript.
Comments 13:Line 163, melon or muskmelon?
Response 13: We sincerely thank your precious suggestion. We have corrected the "muskmelon"to " melon " in the revised manuscript.
Comments14: Figure 2, the legend is incomplete. What does the star, black dots, etc. indicate?
Response 14: We sincerely thank your precious suggestion. We have made corresponding markings and explanations in the annotation of Figure 2. The red star represents the protein in melon. The black circle represents the protein in Arabidopsis. The yellow rectangle represents the protein in rice.
Comments 15:Figure 3, the background color is too dark to read the GeneID. The signature motifs of ALDH family, such as the ones that introduced in the Introduction, should be marked in the right panel. The legend needs to be improved.
Response 15: Thank you for the comment. We have made modifications to Figure 3. Now the gene ID can be read in the revised manuscript.
Comments 16:Line 204-210, I don’t understand.
Response 16: We sincerely thank your precious suggestion. The main idea of this paragraph is: There are 12 collinear relationships between 9 melon ALDH genes and 12 Arabidopsis thaliana ALDH genes. We have made corresponding modifications to this paragraph.
Comments17: Figure 4, the legend need to be improved. What does the gray line indicate? What does the scale on the chromosome mean?
Response 17: We sincerely thank your precious suggestion. The gray lines represent the orthologous relationships of the genes between melon , Arabidopsis, and rice. The scale on the chromosome represents the physical location. We have made corresponding markings and explanations in the annotation of Figure 4.
Comments 18:Section 2.6, the authors selected 2 kb upstream of CmALDH gene to conduct the promoter analysis? This relevant information should be provided.
Response 18: Thank you for the comment. We have added relevant information on promoter analysis in the Materials and Methods section in the revised manuscript.
Comments 19:Figure 6 to 8, the statistic analysis results and the raw data for the heat maps should be provided. The value inside each brick is not inconsistent with the color code.
Response 19: Thank you for the comment. We have packaged and uploaded the relevant raw data. The color inside the brick changes according to the FPKM value. The larger the FPKM value, the redder it is. The smaller the value, the bluer the color. When the value is 0, it is white.
Comments 20:Line 284, chilling stress.
Response 20: We sincerely thank your precious suggestion. We have made modifications according to your suggestion in the revised manuscript.
Comments21: Section 2.11, which standard did the authors apply to select the 10 candidates for RT-qPCR analysis? I suggest examining the expression changes of all of the candidates.
Response 21: Dear reviewer, we have performed RT-qPCR on all these 17 ALDH genes. The results show that only the expression levels of these 10 genes are consistent with the trend of transcriptome data. Therefore, we only show these 10 genes. If necessary, I can supplement the result diagrams of the other seven genes.
Comments 22:Line 406, biotic stress.
Response 22: Dear reviewer, I am very sorry. I did not understand your meaning. Line 406 is the annotation part of Figure 10. It refers to the RT-qPCR analysis of melon under salt stress. When you said "biotic stress" here, I did not understand. I hope you can point out my mistake in detail. Thank you very much.
Comments23: Line 481 to 508, the authors recite the Results without any insightful statements. I suggest that they should integrate the results of tissue-specific, developmental and stress responsive transcriptome analysis to conclude the functional model of selected candidate.
Response 23: Respected reviewer, thank you for your careful review and feedback on our article. Regarding the content from lines 481 to 508 that you mentioned, in our discussion, we have tried our best to integrate the results of specific, developmental, and stress response transcriptome analyses. We believe that the current expression has covered the relevant key points. We will continue to pay attention to the suggestions you mentioned and consider how to more comprehensively show the relationship between these results in subsequent studies.
Comments24: Line 511, MELO3C024328?
Response 24: We sincerely thank your precious suggestion. “MELO3C024328” is incorrect content. We have corrected it to “MELO3C025328” in the revised manuscript.
Comments 25:Section 4.6, the methodology of RT-qPCR is incomplete. Such as the chemical regents/kits used for RNA extraction and reverse transcription. Which gene was used as reference gene?
Response 25: Thank you for the comment. We have made additions in Section 4.6 according to your suggestion.
Comments26: Methodology of statistic analysis should be provided.
Response 26: We sincerely thank your precious suggestion. We have made corresponding modifications in Section 4.6. This study was using the 2-△△ct method and processed via Excel 2021.
Additional clarifications
Dear Reviewer,
Greetings! I am truly grateful for the valuable time you've dedicated to reviewing our manuscript and offering insightful suggestions. Your advice has tremendously aided our work. Please accept my sincerest appreciation once more. As this represents my first attempt at independently writing and submitting an academic paper, I acknowledge that certain details may not have been handled appropriately, causing you unnecessary inconvenience. For this, I extend my deepest apologies. May you enjoy good health and success in your endeavors.
Reviewer 2 Report
Comments and Suggestions for Authors
Dear Authors,
This work The article is written quite well. I have a few comments regarding the content:
The bioinformatics part is performed in a standard manner. However, there are caveats to the methodology. It is necessary to indicate in detail in the “materials and methods” what stages of growth and what effects were studied.
qPCR:
1. What kind of device was used.
2. The figure with qPCR data is very poorly formatted. Nothing is clear or visible at all. Revision is impossible
2. Why did you use standard deviation rather than error of the mean?
3. What method was used to statistically process the data?
Best regards
Author Response
Comments1: What kind of device was used.
Response 1: Thank you for pointing this out. We agree with this comment. Therefore, we have made corresponding modifications in Section 4.6 of the manuscript according to your suggestion and clarified the relevant device information.
Comments2: The figure with qPCR data is very poorly formatted. Nothing is clear or visible at all. Revision is impossible
Response 2: Agree. We have We have made major modifications to Figure 10. At present, the relevant data can be clearly seen.
Comments3: Why did you use standard deviation rather than error of the mean?.
Response 3: Mainly to ensure the accuracy and reliability of the data. The use of standard deviation can help set a reasonable fluorescence threshold, thereby more accurately detecting the changes in fluorescence signals in PCR reactions.
Comments4: What method was used to statistically process the data?
Response 4: Thank you for pointing this out. The results obtained from three biological replicates were analyzed using the 2-△△ct method and processed via Excel 2021. We agree with this comment. Therefore, we have made corresponding modifications in Section 4.6 of the manuscript according to your suggestion.
Additional clarifications
Dear Reviewer,
Greetings! I am truly grateful for the valuable time you've dedicated to reviewing our manuscript and offering insightful suggestions. Your advice has tremendously aided our work. Please accept my sincerest appreciation once more. As this represents my first attempt at independently writing and submitting an academic paper, I acknowledge that certain details may not have been handled appropriately, causing you unnecessary inconvenience. For this, I extend my deepest apologies. May you enjoy good health and success in your endeavors.
Round 2
Reviewer 1 Report
Comments and Suggestions for Authors
Thanks for the authors considering my comments and making the revisions. The revised manuscript has been significantly improved and my major concerns have been addressed properly. Below, I outline a few comments and suggestions in response to Authors’ reply.
1. I suggest the authors slightly explain the concept of “aliphatic index” in the main text.
2. Since the ALDH family has been renamed, it would be better to refer them by using ALDH1, etc. instead of the GeneID.
3. I suggest to present the RT-qPCR results of ALL (17) CmALDH genes in figure 10.
4. Comments 22: Line 406, “both abiotic and BIOTIC stresses”
5. Table S1, the GeneID of Actin3
Comments on the Quality of English LanguageMinor editing of English language required.
Author Response
Comments1: I suggest the authors slightly explain the concept of “aliphatic index” in the main text.
Response 1: Thank you for pointing this out. We agree with this comment. Therefore, We have explained the "aliphatic index" in the main text.
Comments2: Since the ALDH family has been renamed, it would be better to refer them by using ALDH1, etc. instead of the GeneID.
Response 2: Agree. We have modified all the GeneIDs in the manuscript according to your suggestion, such as "MELO3C004383” replace with "CmALDH2B1".
Comments3: I suggest to present the RT-qPCR results of ALL (17) CmALDH genes in figure 10.
Response 3: Thank you for your comment. We have conducted qRT-PCR experiments on the remaining 5 genes and analyzed the relevant data. At present, 17 CmALDH genes are all shown in Figure 10.
Comments4: Comments 22: Line 406, “both abiotic and BIOTIC stresses”
Response 4: We sincerely thank your precious suggestion. We have carefully proofread the whole manuscript according to your suggestion.
Comments5: Table S1, the GeneID of Actin3
Response 5: Thank you for the comment. The GeneID of Actin3 is MELO3C008032. We have added the GeneID of Actin3 in Table S1.
Additional clarifications
Thank you for all your professional advice to us. Under your guidance, my manuscript has been greatly improved, and at the same time, I have been greatly improved. Please allow me to express my heartfelt thanks to you. I wish you good health and success in your work!
Reviewer 2 Report
Comments and Suggestions for Authors
Dear Authors,
This work The article is written quite well. I have a few comments regarding the content:
The bioinformatics part is performed in a standard manner. However, there are caveats to the methodology. It is necessary to indicate in detail in the “materials and methods” what stages of growth and what effects were studied.
qPCR:
1. Check italic for genes
2. The captures for figures with expression data is very poorly formatted. Methods and experimental design are unclear
3. Why did you use standard deviation rather than error of the mean? Standard error is often used for statistical analysis
4. Reorganize Fig 10. It is need 285% zoom to see
5. What method was used to statistically process the data? Include in Material and Methods.
6. Professional and scientific proofreading is required
Best regards
Author Response
Comments1: Check italic for genes.
Response 1: Thank you for pointing this out. We agree with this comment. Therefore, We've done a thorough examination of the manuscript, putting all the genes in italics.
Comments2: The captures for figures with expression data is very poorly formatted. Methods and experimental design are unclear.
Response 2: Agree. We have made significant changes to Figure 10. At present, the relevant data can be clearly seen and the experimental methods and designs have been further refined.
Comments3: Why did you use standard deviation rather than error of the mean? Standard error is often used for statistical analysis
Response 3: Thank you for comments. We agree with you very much and have analyzed the data using average error and replotted according to your request.
Comments4: Reorganize Fig 10. It is need 285% zoom to see
Response 4: Thank you for pointing this out. We have rearranged the pictures and made changes according to your comments.
Comments 5:What method was used to statistically process the data? Include in Material and Methods.
Response 5: Thank you for your advice. The results from three biological replicates were analyzed using the 2-ΔΔCt method and processed with Excel 2021. A T-test in SPSS 19.0 was employed to assess the significance of differences in the data, and GraphPad Prism was utilized for plotting. We have added this passage to the Materials and Methods section of the manuscript.
Comments 6:Professional and scientific proofreading is required
Response 6: Thank you for reminding us that we have done a complete proofreading of the manuscript according to your request.
Additional clarifications
Thank you for all your professional advice to us. Under your guidance, my manuscript has been greatly improved, and at the same time, I have been greatly improved. Please allow me to express my heartfelt thanks to you. I wish you good health and success in your work!